# Application of the Fractal Method to the Characterization of Organic Heterogeneities in Shales and Exploration Evaluation of Shale Oil

**Bo Liu [1,2,*] , Liangwen Yao [1,2], Xiaofei Fu [1], Bo He [3] and Longhui Bai [1,2]**

[1]  Institute of Unconventional Oil and Gas, State Key Laboratory Cultivation Base,
    Northeast Petroleum University, Daqing 163318, China; liangwenyao6968@163.com (L.Y.);
    fuxiaofei2008@sohu.com (X.F.); 13504657715@163.com (L.B.)
[2]  State Key Laboratory of Oil and Gas Reservoir Geology and Exploitation,
    Chengdu University of Technology, Chengdu 610051, China
[3]  PetroChina Daqing Oilfield Company, China National Petroleum Corporation, Daqing 163318, China;
    m18831633978@163.com
[*]  Correspondence: liubo6869@163.com or liubo@nepu.edu.cn; Tel.: +86-0459-6504095

**Abstract:** The first member of the Qingshankou Formation, in the Gulong Sag in the northern part of the Songliao Basin, has become an important target for unconventional hydrocarbon exploration. The organic-rich shale within this formation not only provides favorable hydrocarbon source rocks for conventional reservoirs, but also has excellent potential for shale oil exploration due to its thickness, abundant organic matter, the overall mature oil generation state, high hydrocarbon retention, and commonly existing overpressure. Geochemical analyses of the total organic carbon content (TOC) and rock pyrolysis evaluation (Rock-Eval) have allowed for the quantitative evaluation of the organic matter in the shale. However, the organic matter exhibits a highly heterogeneous spatial distribution and its magnitude varies even at the millimeter scale. In addition, quantification of the TOC distribution is significant to the evaluation of shale reservoirs and the estimation of shale oil resources. In this study, well log data was calibrated using the measured TOC of core samples collected from 11 boreholes in the study area; the continuous TOC distribution within the target zone was obtained using the △logR method; the organic heterogeneity of the shale was characterized using multiple fractal models, including the box-counting dimension (Bd), the power law, and the Hurst exponent models. According to the fractal dimension (D) calculation, the vertical distribution of the TOC was extremely homogeneous. The power law calculation indicates that the vertical distribution of the TOC in the first member of the Qingshankou Formation is multi-fractal and highly heterogeneous. The Hurst exponent varies between 0.23 and 0.49. The lower values indicate higher continuity and enrichment of organic matter, while the higher values suggest a more heterogeneous organic matter distribution. Using the average TOC, coefficient of variation (CV), Bd, D, inflection point, and the Hurst exponent as independent variables, the interpolation prediction method was used to evaluate the exploration potential of the study area. The results indicate that the areas containing boreholes B, C, D, F, and I in the western part of the Gulong Sag are the most promising potential exploration areas. In conclusion, the findings of this study are of significant value in predicting favorable exploration zones for unconventional reservoirs.

**Keywords:** fractal; box-counting; Hurst exponent; shale oil; Songliao Basin; China

---

## 1. Introduction

With the growing global energy demand and the success of unconventional shale oil and gas in North America, unconventional reservoirs, an effective substitute for conventional oil-gas resources, have been of great interest in global exploration and exploitation [1]. Although shale is often rich in generated hydrocarbons, the recovery from the shale formation can be hard due to permeability, which makes shale oil-gas exploration quite difficult. Due to the restrictions of lake basins, limited water circulation, and the influences of climate and local detritus sources, continental shale deposits exhibit fast lateral lithofacies changes [2]. Current exploration practices and studies indicate that the total organic carbon content (TOC) is one of the most important indicators of oil enrichment within the shale formations [3]. The spatial distribution of the TOC can be highly heterogeneous and can vary in magnitude, even at the millimeter scale. Geologists have diligently tried to develop a means of effectively characterizing the TOC heterogeneity, rather than simply identifying the organic matter abundance of a borehole using the average TOC [4]. It is very important to determine a shale formation's organic heterogeneity when evaluating shale oil-gas reservoirs [5].

Combining mathematical theory with geological data remains the key to characterizing the organic heterogeneity of a shale formation and to evaluating its exploration potential. Fractal theory is a non-linear mathematical method developed by Mandelbrot (1983) [6] that can be used to calculate the resource potential of a variety of mineral deposits and to predict potential exploration targets when it is combined with geochemical data [7,8]. For example, the vertical distribution of the content of a given element obeys a power law, which is consistent with the fractal model [9,10]. These studies successfully investigate the fractal models to characterize the distribution of geochemical elements in boreholes, which is of significance to evaluate the quality and quantity of mineral resources. Fractal dimensions, including box dimension, power-law frequency, and Hurst exponents, were widely used to evaluate the datasets based on boreholes.

In this study, we characterize the vertical TOC distribution of the shale in the first member of the Qingshankou Formation, in the Gulong Sag in the northern part of the Songliao Basin, using multiple fractal models, including the box dimension model, the power law model, and the Hurst exponent model. We also investigate the shale's horizontal heterogeneity using inverse distance weighting (IDW) and predict the distribution of the shale oil exploration targets within the study area using multiple indices.

## 2. Mathematical Methods

### 2.1. Box Dimension Method

The box dimension (Minkowski dimension) method is extensively used to calculate the fractal dimension of a data set (a set in the Euclidean space). This method uses square boxes with a certain edge length to cover all of the data points, then the edge length is gradually reduced, and finally, the limit of the ratio between the minimum number of boxes required and the edge length is calculated [7,11]. In this study, the box dimension was calculated using Matlab. This involved covering the data points using grids ($\delta$) with different sizes, counting the minimum numbers of grids required ($n(\delta)$), projecting the values of $\delta$ and $n(\delta)$ onto the double logarithmic coordinate system, and conducting regression fitting using the least squares method to calculate the regression coefficient and the autocorrelation coefficient $R^2$. Formally, the box dimension Bd is given by the limit (1).

$$\mathrm{Bd} := \lim_{\delta \to 0} \frac{\log N(\delta)}{\log 1/\delta},$$ (1)

Roughly speaking, it means that the dimension is the exponent Bd of the power-law (2).

$$N(\delta) \propto \delta^{-\mathrm{Bd}}.$$ (2)

here δ is the edge length of the box, N(δ) is the minimum number of required boxes; δ and n(δ) were plotted on a double logarithmic coordinate system. A straight line suggesting a negative correlation was fitted using the least squares method. Bd was determined from the slope of the line (−Bd). The Bd of a one-dimensional borehole profile in a two-dimensional space is described by Equation (2).

## 2.2. Power-Law Frequency Model

This Power law model, which is usually used to represent the frequency distribution characteristics of dates, has been applied in a variety of fields [9,10,12]. Moreover, it is also used to determine an anomaly's lower limit [9]. It can be written as

$$N(\delta) \propto \delta^{-Pd} \tag{3}$$

Or

$$\log[N(\delta)] = C - Pd \log(\delta) \tag{4}$$

here δ is the characteristic scale, N(δ) is the amount of the data ≥δ, C is a constant, and *Pd* is the fractal dimension. The scatter plot of δ and n(δ) can be divided into two segments that are separated by the inflection point. The *Pd* of each segment was calculated using piecewise fitting.

## 2.3. Return-to-Scale Model

Return-to-scale (R/S) analysis can be used to calculate the Hurst exponent [13]. This method was developed by Hurst (1951) [14]. The Hurst exponent was initially used to investigate hydrolysis in the Nile Valley, and it has since been perfected through numerous modifications [15]. In the R/S model, the range (*R*) and standard deviation (*S*) corresponding to different sub-series are obtained using a given time sequence after calculating the difference sequence that represents the growth rate or attenuation rate. Based on this, the R/S ratio is obtained. If the R/S ratio exhibits a power-law distribution for each sub-series, the power exponent is the Hurst exponent. In this method, the cumulative deviation *X(i)* is defined as

$$X(i,n) = \sum_{i=1}^{N} \left( X(i) - \overline{X}_n \right), \tag{5}$$

where

$$\overline{X}_n = \frac{1}{n} \sum_{i=1}^{n} X(i). \tag{6}$$

The range is defined as

$$R(n) = max(X(i,n)) - min(X(i,n)), 1 \le i \le n. \tag{7}$$

The standard deviation is defined as

$$S(n) = \sqrt{\frac{1}{n} \sum_{i=1}^{n} \left( X(i) - \overline{X}_n \right)}. \tag{8}$$

The range is normalized by dividing the range by the standard deviation, which eliminates the influence of the dimension, and the ratio of the range and standard deviation is obtained as follows:

$$\frac{R}{S} = \frac{R(n)}{S(n)}. \tag{9}$$

Previous studies have found that the R/S ratio conforms to the following empirical formula for some data points [16]:

$$\frac{R}{S} \propto n^{H}, \tag{10}$$

where $i = 1, 2, 3, \ldots$ ; $n$ is an arbitrary integer; $X(i)$ is the cumulative deviation; $\overline{X}$ is the mean of the sequence; $R$ is the range; $S$ is the standard deviation; and $H$ is the Hurst exponent, which ranges from 0 to 1.

The data sequence is a random sequence (such as white noise) when the Hurst exponent is 0.5; it is a negatively correlated sequence when the Hurst exponent is less than 0.5, and it is a positively correlated non-random sequence when the Hurst exponent is greater than 0.5. For a non-random data sequence, the data that is within an upward trend or within a downward trend within a certain time sequence will remain in that trend throughout the interval. Normally, on a one-dimensional profile, the relationship $H = D - 2$ exists between the fractal dimension ($D$) and the Hurst exponent ($H$). On a one-dimensional profile of a two-dimensional space, the possible range of the fractal dimension is 1–2, and that of $H$ is 0–1. The Hurst exponent can be obtained from the linear regression of $R/S$ verses n in the double logarithmic coordinate system [10].

### 2.4. Interpolation Method

Based on the fractal analyses, the contour map of fractal parameters is obtained via the kriging and inverse distance weighting (IDW) interpolation in ArcGIS version 10. The IDW parameters used had a weighting power of 2 and a minimum neighbors of 10.

### 3. Geologic Setting

The Songliao Basin is located in northeastern China. It has an area of approximately $26 \times 10^4$ km$^2$. It can be divided into six first-order tectonic units based on the regional structural characteristics and evolutionary characteristics of the basin: the central depression, the northern plunge, the western slope, the southwestern uplift, the northeastern uplift, and the southeastern uplift (Figure 1a). The Gulong Sag is located in the western part of the central depression to the west of the Daqing placanticline and to the east of the Longhupao and Honggang terraces. The Gulong Sag has an area of approximately 3700 km$^2$ (Figure 1b). It is a monoclinal structure that is high in the northwest and low in the southeast; it exhibits favorable succession; the dip angle within the sag varies gradually, and the structural patterns of the shallow and deep strata are similar [17].

In the Late Cretaceous, the Songliao Basin was a large continental depression. The dark mudstone of the Qingshankou Formation was deposited throughout a large area of the basin. The Qingshankou Formation is divided into the first, second, and third members from base to top. The lake's area was approximately $10^5$ km$^2$ and the maximum thickness of the mudstone is 129.6 m (61.5 m on average) in the first member of the Qingshankou Formation (Figure 1c,d). The shale's organic matter originated from lacustrine alginate. The organic matter is mainly Type-I and Type-II1, with a high organic matter enrichment of greater than 1%. The shale in this area is mostly in the mature stage [4]. The Gulong Sag is a deep-water lake basin depression that exhibits long-term succession of dark fine-grained sediments. During deposition of the Qingshankou Formation, the Yingtai paleo-drainage system developed in the western part of the depression. Sediments from the delta front had been deposited into the basin due to gravitational instabilities. A great volume of detritus was transported, forming turbidite sand bodies within the shale intervals, which increased the heterogeneity of these shale intervals [18].

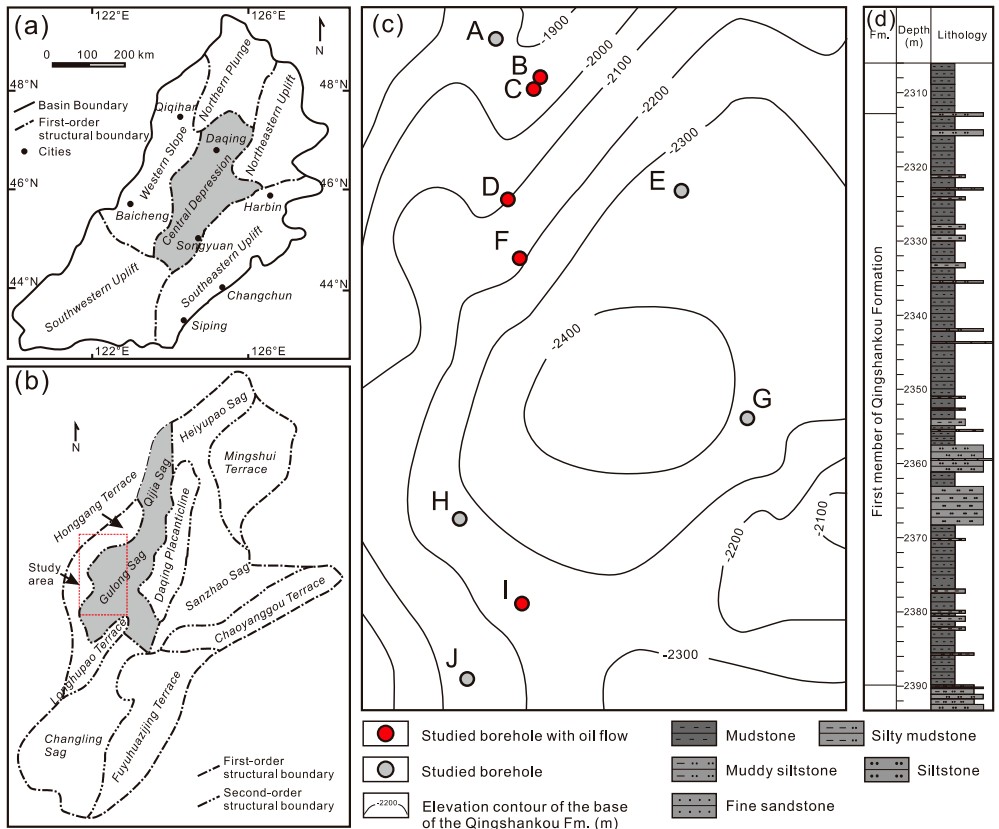

**Figure 1.** Generalized map showing the division of the tectonic units in (**a**) the Songliao Basin and (**b**) the Central Depression. (**c**) Base contour map of the first member of the Qingshankou Formation ($K_2qn^1$) in the Gulong Sag. (**d**) Lithologic sketch of $K_2qn^1$.

## 4. Sampling and Analysis

In this study, 47 shale samples from the first member of the Qingshankou Formation were collected from 10 exploratory boreholes in the Gulong Sag (Figure 1c). These samples were crushed, and the carbonate minerals were removed using dilute hydrochloric acid. Then, the TOC of the samples was determined using an Eltra Helios C elemental analyzer [4]. The well log data was demarcated based on the measured TOC data. The vertical distribution of the TOC of the samples from the 10 boreholes in the first member of the Qingshankou Formation was calculated using the $\Delta\log R$ logging hydrocarbon source rock evaluation method. These calculations were used to create a continuous TOC distribution profile with an interval of 1 m (Figure 2).

The $\Delta\log R$ method was proposed by Passey et al. (1990) [19]. The equation for calculating the TOC is as follows:

$$\Delta\log R = \log(R/R_{baseline\ n}) + 0.02(\Delta t - \Delta t_{baseline\ n}), \tag{11}$$

$$TOC = 10^{(2.297\ -\ 0.1688LOM)}\ \Delta\log R + \Delta TOC, \tag{12}$$

where $R$ is resistivity log ($\Omega$·m); $\Delta t$ is transit time difference log ($\mu$s/ft); TOC is the total organic carbon content (%); and the Labile organic matter (LOM) is an organic matter thermal metamorphism index, which reflects the maturity of the hydrocarbon source rock. We made the resistivity log and the transit time difference log of the fine-grained non-source rocks coincide as a line, which is the called base line (the corresponding electrical resistivity and transit time difference are $R_{baseline\ n}$ and $\Delta t_{baseline\ n}$, respectively). The distance between these two logs is the difference in amplitude ($\Delta\log R$). For the hydrocarbon source rock, the TOC is directly proportional to $\Delta\log R$ and is inversely proportional to the maturity (LOM). In addition, the background TOC value ($\Delta TOC$), which is unidentifiable on the logging curves, should also be considered in practical applications.

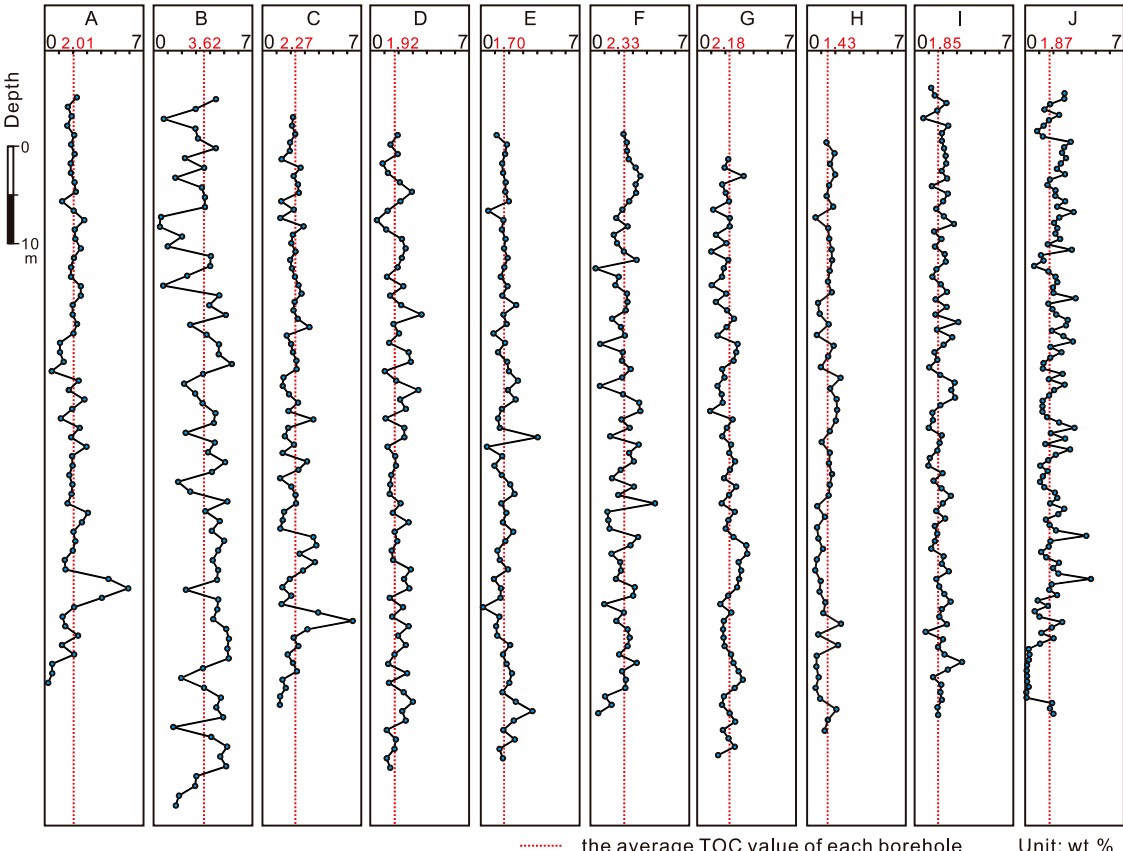

**Figure 2.** Vertical distribution of the total organic carbon (TOC) of selected boreholes in the Gulong Sag (The distance between the two TOC values is 1 m).

## 5. Calculation Results

### 5.1. Bd of the TOC Distribution

In nature, fractals are self-similar and can be quantified using a fractal dimension. Researchers have developed a variety of methods to calculate fractal dimensions [10]. The Bd method is the most widely used across various fields since its mathematical calculation and empirical estimation are relatively simple. The Bd method can be used to describe the distribution of the TOC aggregation. A higher Bd value indicates a concentrated TOC distribution, while a lower value indicates a dispersed TOC distribution [20]. In this study, the various Bds of the various bore holes (Figure 3, Table 1) indicate that Bd ranges from 0.96 to 1.04 and the correlation coefficients are greater than 0.96. Even larger Bd values were found in boreholes where oil has been discovered within shale reservoirs, corresponding to short secondary migration.

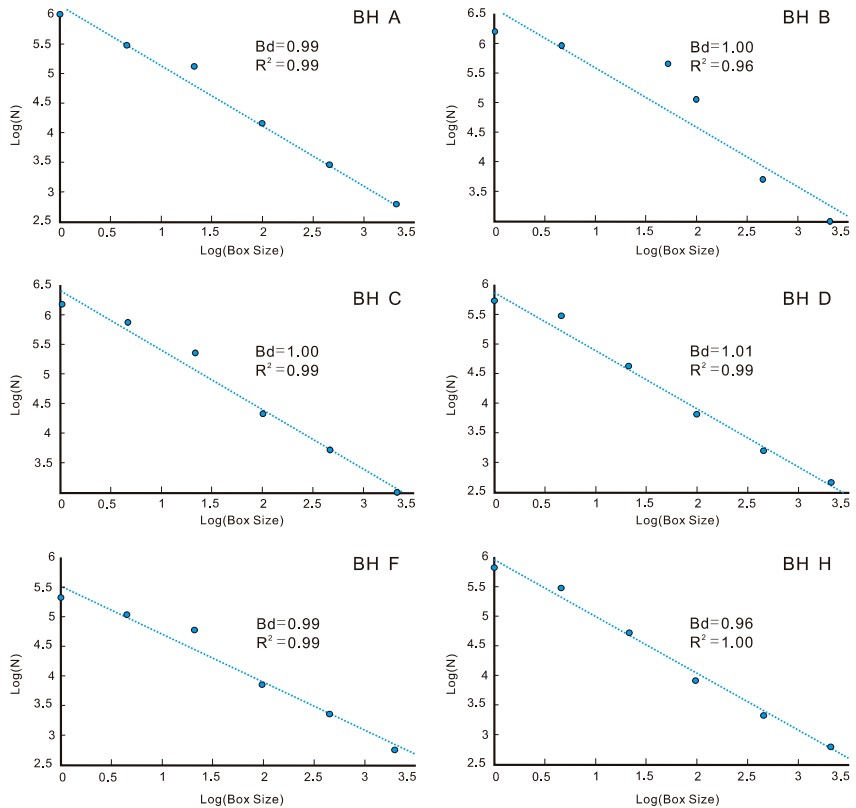

**Figure 3.** Log–log plots of box size versus number.

**Table 1.** Fractal parameters of the Gulong Sag.

| BH | Number of Samples | Min (wt.%) | Max (wt.%) | Average (wt.%) | Std | CV |
|----|-------------------|-----------|-----------|----------------|-----|-----|
| A | 63 | 0.24 | 5.91 | 2.01 | 0.87 | 0.44 |
| B | 73 | 0.46 | 5.57 | 3.62 | 1.34 | 0.37 |
| C | 71 | 1.25 | 6.41 | 2.27 | 0.83 | 0.36 |
| D | 74 | 0.53 | 3.46 | 1.69 | 0.50 | 0.30 |
| E | 67 | 0.15 | 4.04 | 1.70 | 0.62 | 0.36 |
| F | 70 | 0.38 | 4.60 | 2.33 | 0.81 | 0.35 |
| G | 72 | 0.91 | 3.54 | 2.18 | 0.57 | 0.26 |
| H | 56 | 0.30 | 2.89 | 1.43 | 0.73 | 0.51 |
| I | 84 | 0.61 | 3.36 | 1.85 | 0.52 | 0.28 |
| J | 116 | 0.07 | 4.64 | 1.87 | 0.89 | 0.47 |

| Box Dimensions | | | Power-Law Frequency | | | | | Hurst Exponent | | |
|----|-----------|-------|-------------|-------|--------------------------|-------------|-------|-------|------|-------|
| BH | Dimension | $R^2$ | Dimension 1 | $R^2$ | Inflection Point (wt.%) | Dimension 2 | $R^2$ | Hurst | D | $R^2$ |
| A | 0.99 | 0.99 | 0.13 | 0.78 | 1.69 | 4.94 | 0.95 | 0.49 | 1.51 | 0.86 |
| B | 1.00 | 0.96 | 0.28 | 0.75 | 3.89 | 8.37 | 0.85 | 0.24 | 1.76 | 0.73 |
| C | 1.00 | 0.99 | 0.84 | 2.26 | 3.62 | 8.40 | 0.97 | 0.31 | 1.69 | 0.80 |
| D | 0.98 | 0.99 | 0.50 | 0.70 | 1.60 | 4.95 | 0.99 | 0.32 | 1.68 | 0.77 |
| E | 0.98 | 1.00 | 0.11 | 0.58 | 1.40 | 4.90 | 0.93 | 0.37 | 1.63 | 0.92 |
| F | 1.04 | 0.98 | 0.24 | 0.77 | 2.16 | 5.41 | 0.87 | 0.23 | 1.77 | 0.71 |
| G | 1.01 | 0.99 | 0.81 | 0.73 | 2.20 | 6.35 | 0.95 | 0.36 | 1.64 | 0.92 |
| H | 0.96 | 1.00 | 0.40 | 0.98 | 1.61 | 4.59 | 0.94 | 0.39 | 1.61 | 0.80 |
| I | 1.02 | 1.00 | 0.82 | 0.78 | 1.92 | 6.13 | 0.98 | 0.26 | 1.74 | 0.83 |
| J | 1.02 | 0.99 | 0.11 | 0.70 | 1.63 | 3.71 | 0.94 | 0.31 | 1.69 | 0.75 |

Abbreviations: BH = borehole, CV = coefficients of variance, $R^2$ = the degree of fitting, D = dimension.

### 5.2. Power-Law Distribution of TOC

The fractal dimension (Pd) obtained using the power law model is consistent with the distribution frequency of the relatively high TOC value. The double logarithmic coordinate plots of the TOC contents versus the summed cumulative number of samples with ordered TOC data sequence (Figure 4) indicate that the vertical TOC distribution exhibits multi-fractal features. The high value of the anomaly's lower limit, i.e., the large inflection point value, suggests a greater deviation of the TOC from the average value and a higher concentration of organic matter. Dimension 2 is the fractal dimension when the TOC is greater than the anomaly's lower limit. A larger fractal dimension indicates a higher concentration of TOC values and a slow vertical change in TOC. Similarly, the coefficient of variation (CV) indicates the amplitude of the vertical variation in TOC. A smaller CV indicates a slow change in the TOC, i.e., high homogeneity of TOC values. Figure 5 shows the regression line between the fractal dimension (D) and the coefficient of variation (CV) and the negative correlation between the fractal dimension of TOC and CV.

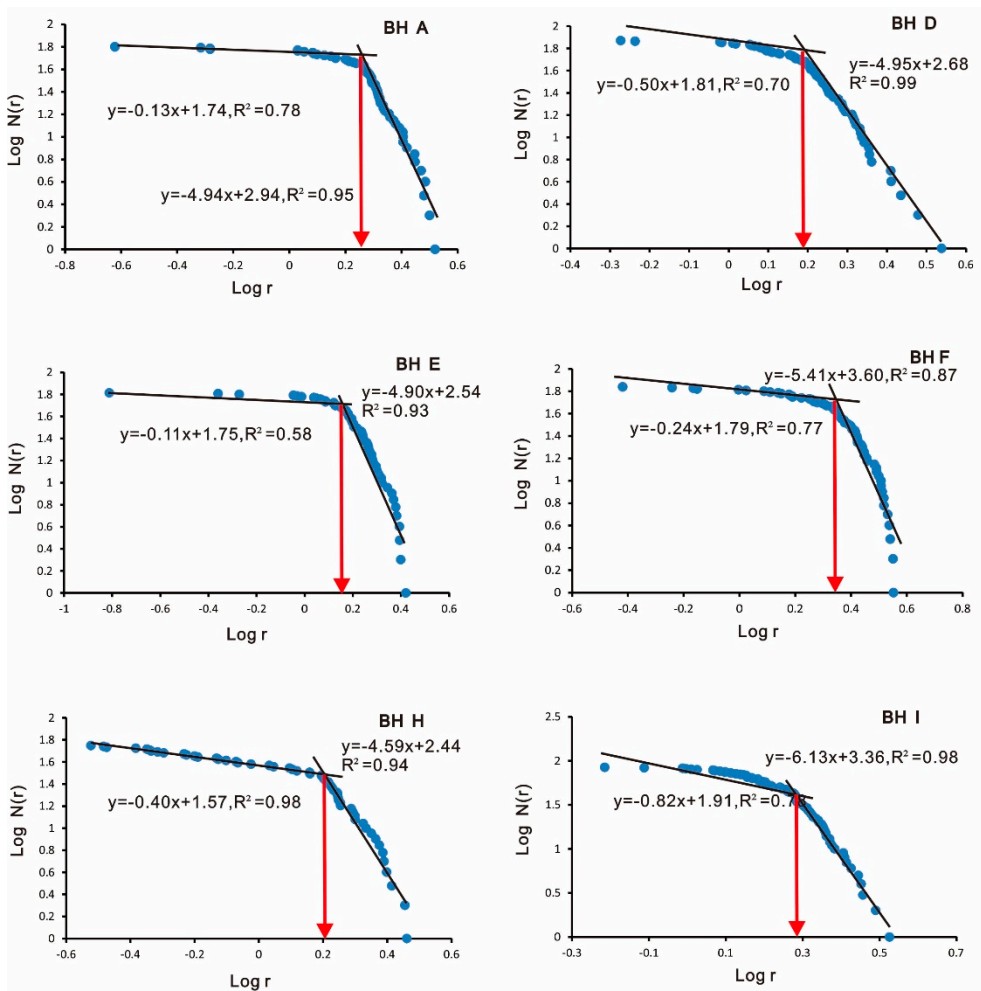

**Figure 4.** Log–log plots of the cumulative number of samples versus total organic carbon.

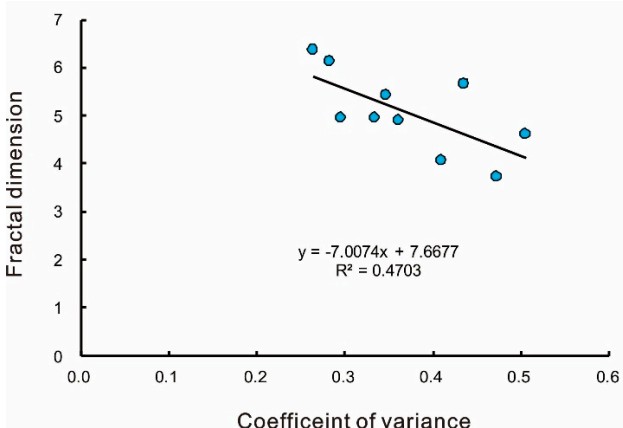

**Figure 5.** Relationship between the fractal dimension and the coefficient of variation.

### 5.3. Hurst Exponent

The Hurst exponent of the TOC was calculated using the R/S model. The Hurst exponent varied from 0.23–0.49, and the degree of fitting ($R^2$) was greater than 0.71. The Hurst exponent can be used to quantitatively describe the long-range dependence, i.e., the continuity. As shown by the double logarithmic coordinate graph (Figure 6), the vertically-distributed Hurst exponents of the TOC of the various boreholes in the study area are all less than 0.5. The closer the Hurst exponent is to 0, the less randomness and the higher the correlation. Therefore, the Hurst exponent can also be used to indicate the hydrocarbon generation potential. As is expected, the TOC correlation is higher and the Hurst exponent is smaller in regions with a higher hydrocarbon generation potential.

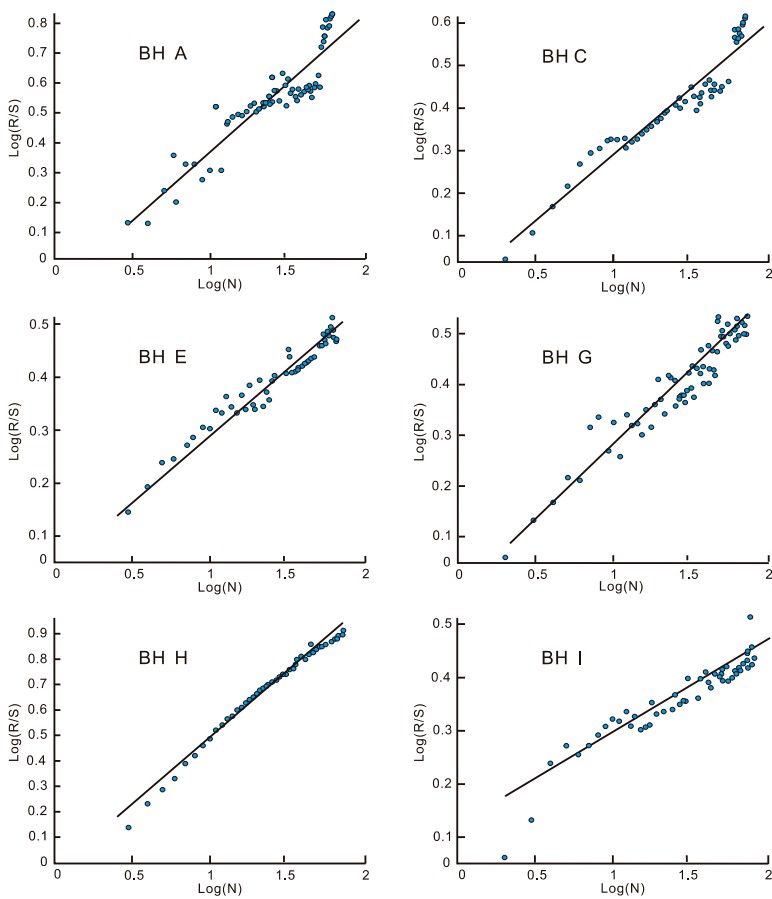

**Figure 6.** Log–log plots of R/S versus the number of samples (N).

## 6. Prediction of a Potential Exploration Area

In this study, the average TOC, CV, Bd, Hurst exponent, and fractal dimension of each borehole were calculated. A low Hurst exponent (equivalent to a high D value), high average TOC, high Bd, and high inflection point value usually represent a high organic matter abundance and a favorable hydrocarbon generation potential, whereas a high Hurst exponent (equivalent to a low D value), low CV, and low inflection point indicate a discontinuous organic matter distribution and a poor hydrocarbon generation potential. Therefore, the average TOC, CV, Bd, D, Hurst exponent, and inflection point value can be used to identify favorable exploration areas (Figure 7).

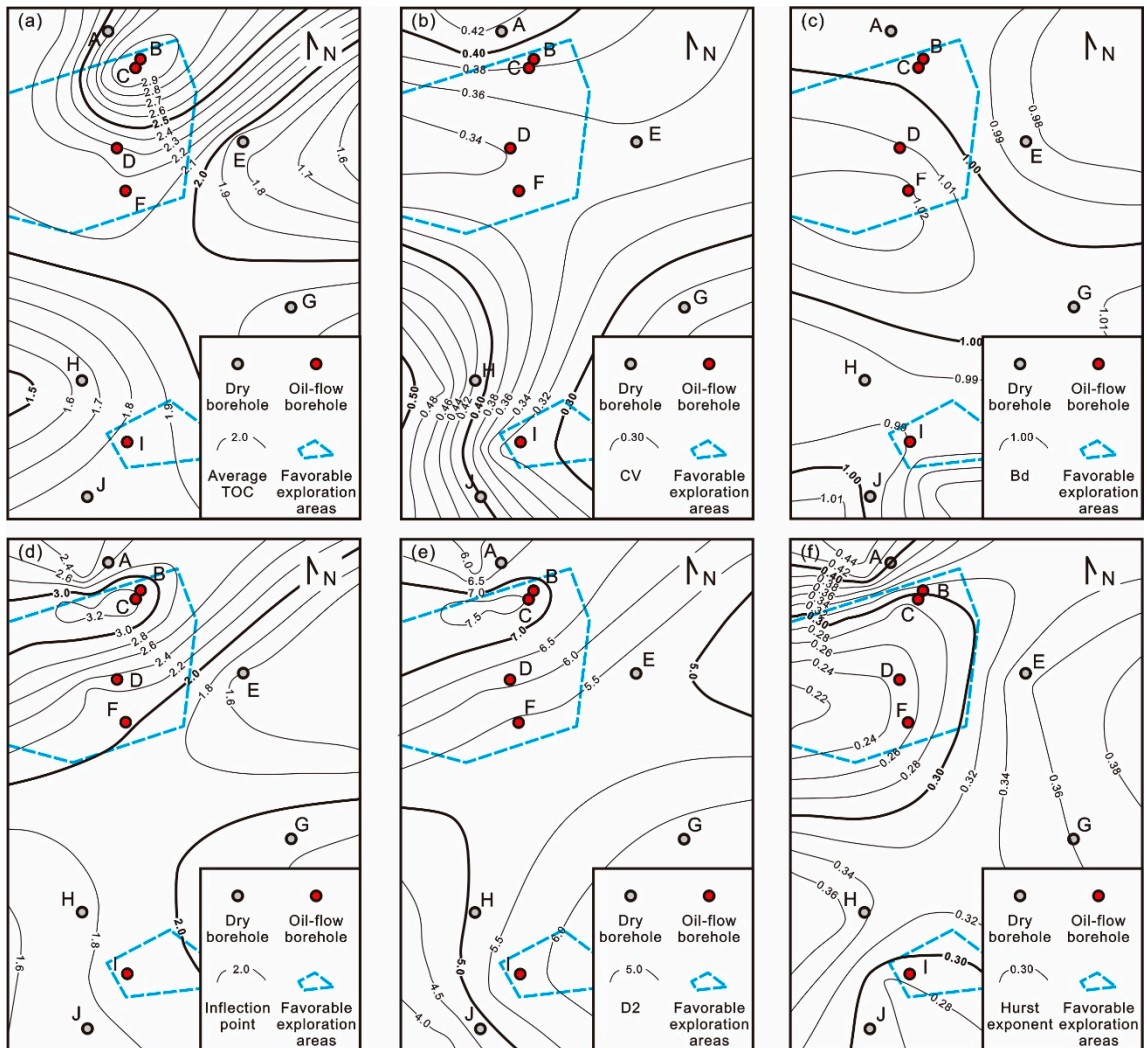

**Figure 7.** Predictive map of total organic carbon (TOC) exploration targets created based on statistical analysis and fractal modeling: (**a**) average TOC grades; (**b**) coefficients of variance (CV); (**c**) box dimensions (Bd); (**d**) inflection point of the bi-fractal; (**e**) fractal dimension 2 of the bi-fractal; and (**f**) Hurst exponents. Favorable areas marked in blue fit the threshold of fractal parameters.

To demonstrate the reliability of the fractal model in identifying potential exploration targets, the parameters of the various boreholes in the study area were calculated. The parameters of the non-drilled areas were obtained by interpolation using IDW in ArcGIS. The results indicate that the area containing boreholes C, B, D, F, and I are located in the major potential exploration areas, which is consistent with the exploration results.

Based on the test results, boreholes were divided into two groups, oil-flow and dry boreholes (different color in Figure 7). The average TOC and fractal parameters of these two groups were

compared to get the threshold value of favorable shale oil exploration target (Figure 8). Bd and average TOC show only moderate trends to separate two groups (1.0 and 1.8, respectively), showing the limitation of conventional parameters. Oil-flow groups have obvious higher inflection points (>1.9) and higher values for dimension 2 (>5.4), while also having lower Hurst exponents (<0.31) and CVs (<0.37), demonstrating effectiveness as methods to predict the potential exploration target.

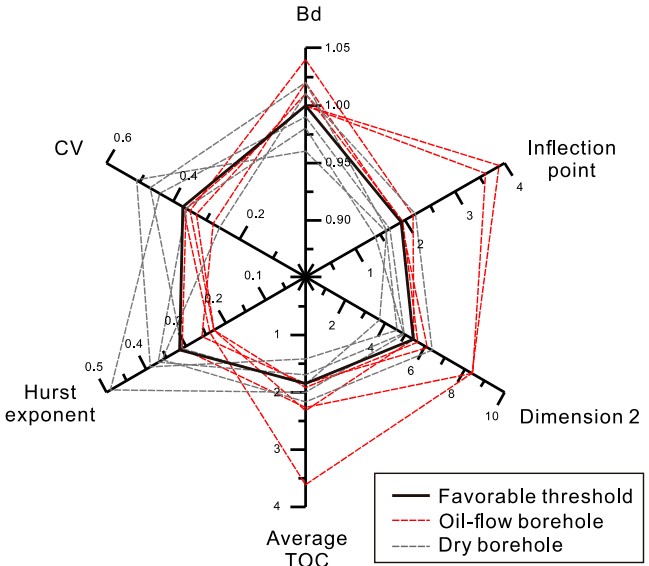

**Figure 8.** Radar plot of conventional parameter average TOC and fractal parameters of boreholes. TOC: total organic carbon. CV: coefficients of variance. Bd: box dimensions.

## 7. Conclusions

(1) Vertical distribution of TOC values can be characterized using fractal models to evaluate the exploration potential and predict the favorable area of shale oil, especially in sparsely-drilled areas during the initial stage of exploration.

(2) In this study, Bd varied between 0.96 and 1.04, which suggests the self-similarity of the vertical TOC distribution of the boreholes. A larger Bd indicates a higher average organic matter abundance. The Power law model indicates that the vertical TOC distribution exhibits multi-fractal features. That is, the larger the fractal dimension is, the higher the proportion of relatively high TOC values; and the larger the inflection point value (anomaly's lower limit) is, the greater the hydrocarbon generation potential of the shale. Low Hurst exponents indicate favorable exploration areas. The Hurst exponents of the boreholes in the study area with high TOC are less than 0.5, suggesting that the organic matter is continuously vertically distributed.

(3) Favorable exploration areas were predicted using interpolation maps of the mean TOC, CV, Bd, inflection point value, fractal dimension, and Hurst exponent. The exploration potential of this region can be initially identified using the mean TOC and Bd, while the distribution of favorable exploration areas can be predicted by high inflection point values (>1.9), high fractal dimensions (>5.4), low CV (<0.37), and low Hurst exponents (<0.31).

**Author Contributions:** B.L. conceived and wrote the paper draft. L.Y. performed the calculation. B.L., B.H. and L.B. contributed to the analysis and discussion of the results. X.F. revised the paper.

**Funding:** This study was supported by Open Fund (PLC20180402) of the State Key Laboratory of Oil and Gas Reservoir Geology and Exploitation (Chengdu University of Technology) and the National Natural Science Foundation of China (No. U1562214). We would like to thank the University Nursing Program for Young Scholars with Creative Talents in Heilongjiang Province (UNPYSCT-2015077), and the National Basic Research Program of China (2016ZX05003-002) for financial support.

**Acknowledgments:** We are also grateful to the Editor and two anonymous reviewers for their suggestions and comments, which significantly improved the quality of the manuscript.

**Conflicts of Interest:** The authors declare no conflict of interest.

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
