# Peer review of "Application of the Fractal Method to the Characterization of Organic Heterogeneities in Shales and Exploration Evaluation of Shale Oil"

_jmse, doi:10.3390/jmse7040088_

Reviewer 1 Report

This research paper deals with the characterization of  the vertical TOC distribution of the shale in the first member of the Qingshankou Formation, in the Gulong Sag in the northern part of the Songliao Basin, using  multiple fractal model. The paper is written well and contains interesting evaluation for TOC.  This paper can be considered for publication after implementing following comments

In the introduction author stated ‘’ For example, the vertical 60 distribution of the content of a given element obeys a power law, which is consistent with the fractal 61 model (Zuo et al., 2009; Nazarpour et al., 2015)’’ would you please elaborate more on this and present more details about these two sources.

Non of these topics covered well in introduction nor in methods section ‘’including the box dimension model, the power law model, and the Hurst exponent model.

Authors said ‘’ We also investigate the shale’s horizontal heterogeneity using inverse distance weighting (IDW) and predict the distribution of the shale oil exploration targets within the study area using multiple indices’’ If IDW is your main method then

Section’’ 4. M athematical methods’’ should be replaced by section 2. Please note that methods should be described ahead of geological setting.

There are some mismatch between results presented in abstract and conclusion , please check and correct

Author Response

Dear Editor and reviewers,

Thank you very much for your letter and for the reviewer’s comments concerning our manuscript. I had a close look and revised portions of the text which are marked in red in the paper.

Reviewer 2 Report

This is a very relevant paper on identifying target for oil shale exploration, based on standard geochemical analyses and mathematical models. The scientific content is very well presented. The write up can be improved by a short abstract/table/figure briefly simplifying the mathematical models used, for the understanding of scientists working in the field of oil shale but not necessarily with a core mathematics background. Further some scope of this work for future, when integrated with other parameters such as thermal maturity can be indicated.

Author Response

(The authors gave the same response as above.)

Reviewer 3 Report

Dear authors,

Please find my comments in attached PDF. My general impression is that you better developed mathematical part of your paper than geological one. It would need to be more balanced.

Author Response

Thank you very much for your comments with detailed marks in the manuscript. We have revised the whole manuscript carefully following your suggestion. Key notes are listed below.

1、The name of the member? Not first, but the youngest or the oldest.

An: The name of the first member customarily means the oldest member in many previous studies. To avoid confusion, we kept the name of the first member, but added the description in the section of Geological Setting “The Qingshankou Formation is divided into the first, second and third members from base to top”.

2、Figure 2 was added a depth scale.

3、A new figure and a paragraph were added in the final section to support the statement with some numbers.

4、Figure 7 was revised to have equidistance, arrow to the north, legend and favorable exploration areas.

5、Conclusions one was re-written.

Round  2

Reviewer 3 Report

Dear author,

I accepted your manuscript.

Just two final remarks for galley-proofs:

1. iThebticate showed the 5 % similarity with paper:

Application of fractal models to characterization and evaluation of vertical distribution of geochemical data in Zarshuran gold deposit, NW Iran (Nazarpoura et al.). Mostly in the theoretical part. Would be wise to change word order in some references.

2. Map of Fig. 7. Nice, but still are missing some obligatory mapping elements - equidistance value, north pointer, type of contour line in legend. 

But you are still NOT using the standard equidistances, e.g. (a) 1.28, 1.61, 1.95 are used and appropriate is 1.0, 1.5, 2.0 etc. etc.

If you wish to send impression that you know what you do with mapping, I recommend to correct isolines (it do not change results nor conclusions).

With regards,

Tomislav Malvić, Univ. of Zagreb

Author Response

Dear Editor,

Thank you for your kind suggestions.

We have revised the theoretical part of the manuscript.

The figure 7 has added the north pointer and type of contour line in legend in each one. For the limited interval of values, it is really hard to make every equidistance integral number. However, we make equidistance values are 0.1, 0.02, 0.01, 0.2, 0.5, and 0.02 in each map respectively, based on the range of different parameter values. And each 5th contour is bolder.

We hope the revised manuscript is ready to accept. If you have any questions, please contact us. I will do my best to make this work better. Thank you for your effort and for considering our manuscript for publication.

Once again, thank you very much for your comments and suggestions.

Sincerely

Dr. Bo Liu